# Temperature stress induces mites to help their carrion beetle hosts by eliminating rival blowflies

**Syuan-Jyun Sun[1,2]\*, Rebecca M Kilner[1]**

[1]Department of Zoology, University of Cambridge, Cambridge, United Kingdom; [2]Institute of Ecology and Evolutionary Biology, National Taiwan University, Taipei, Taiwan

**Abstract** Ecological conditions are known to change the expression of mutualisms though the causal agents driving such changes remain poorly understood. Here we show that temperature stress modulates the harm threatened by a common enemy, and thereby induces a phoretic mite to become a protective mutualist. Our experiments focus on the interactions between the burying beetle *Nicrophorus vespilloides*, an associated mite species *Poecilochirus carabi* and their common enemy, blowflies, when all three species reproduce on the same small vertebrate carrion. We show that mites compete with beetle larvae for food in the absence of blowflies, and reduce beetle reproductive success. However, when blowflies breed on the carrion too, mites enhance beetle reproductive success by eating blowfly eggs. High densities of mites are especially effective at promoting beetle reproductive success at higher and lower natural ranges in temperature, when blowfly larvae are more potent rivals for the limited resources on the carcass.

\*For correspondence:
sjs243@ntu.edu.tw

**Competing interests:** The authors declare that no competing interests exist.

## Introduction

Protective mutualisms among macro-organisms are both widespread and well-known (*Clay, 2014*; *Palmer et al., 2015*; *Hopkins et al., 2017*). They involve one species defending another species from attack by a third party species, in exchange for some form of reward (*Clay, 2014*; *Palmer et al., 2015*; *Hopkins et al., 2017*). Theoretical analyses predict that mutualisms like this can evolve when a commensal or mildly parasitic species, that lives in or upon its host, is induced to become a protective mutualist upon exposure to an environmental stressor (*Fellous and Salvaudon, 2009*; *Lively et al., 2005*; *Hopkins et al., 2017*; *Rafaluk-Mohr et al., 2018*). The stressor can be biotic (*Ashby and King, 2017*; *Clay, 2014*; *Ewald, 1987*; *Lively et al., 2005*; *Schwarz and Müller, 1992*) or abiotic (*Corbin et al., 2017*; *Engl et al., 2018*; *Hoang et al., 2019*).

Although the adaptive evolution of mutualisms has been studied in detail, the contextual factors that drive equivalent variation in the expression of mutualisms on an ecological timescale are relatively less well understood (*Chamberlain et al., 2014*; *Jaenike et al., 2010*; *Hoeksema and Bruna, 2015*), especially for protective mutualisms (*Hopkins et al., 2017*; *Palmer et al., 2015*). In particular, it is unclear how different biotic and abiotic factors combine to influence the expression of a mutualism, especially when conditions vary locally. Nor is it well understood whether the extent of mutualism is density-dependent (*Hoeksema and Bruna, 2015*; *Palmer et al., 2015*). Here we investigate how biotic and abiotic stressors combine to induce the context-dependent expression of a protective mutualism. Specifically, we determine how temperature and partner density interact with the presence of a third party enemy species to influence the likelihood that a phoretic organism can be induced within a single generation to become a protective mutualist.

Our experiments focus on burying beetles (*Nicrophorus vespilloides*), which use the dead body of a small vertebrate to breed upon (*Scott, 1998*). A pair of beetles works together to convert the

carcass into an edible carrion nest for their larvae by removing any fur or feathers, and rolling the meat into a ball. The beetles also reduce competition with rival species for the resources on the dead body by smearing the flesh in antimicrobial exudates, consuming eggs laid by rival insects and concealing the body below ground (*Chen et al., 2020*; *Duarte et al., 2018*; *Scott, 1998*). During carcass preparation, beetle eggs are laid in the surrounding soil and then hatch within 3–4 days. The larvae crawl to the carcass and feed themselves on the edible nest, where they are also fed and defended by both parents. Within a week of hatching, the larvae disperse away from the scant remains of the carcass to pupate, while adults fly off – often to breed again.

Adult burying beetles carry up to 14 species of mites, which also breed on carrion and which use the burying beetle as a means of transport between breeding opportunities. The *Poecilochirus carabi* species complex is the most salient and common of these mite species (e.g. *Wilson, 1983*; *Schwarz et al., 1998*), and it is the focus of this study. *P. carabi* travels as sexually immature deutonymphs on the burying beetle, and derives no nourishment directly from its host while it is on board (*Wilson and Knollenberg, 1987*). Upon arrival at a carcass, the deutonymphs alight and moult into adults, which then reproduce. The next generation of mite deutonymphs is ready to disperse by the time the adult burying beetles cease caring for larvae and leave the breeding event. Roughly 90% of deutonymphs disperse on the departing adults rather than on the burying beetle's larvae (*Schwarz and Müller, 1992*).

*P. carabi* is often described as a phoretic mite because it uses burying beetles (*Nicrophorus* spp.) to travel between breeding opportunities on carrion, and seemingly imposes few costs on its hosts during transportation. Phoretic interactions are thought to pave the way for further interactions between host and phoront that have more positive or negative effects on host fitness. This is especially likely when interactions between host and phoront endure beyond the transport phase (*White et al., 2017*). For example, female *Trichogramma* parasitoid wasps hitch a relatively cost-free ride to their butterfly hosts' egg-laying site, but upon arrival are easily able to locate butterfly eggs to parasitise (*Fatouros and Huigens, 2012*). Likewise, the phoretic mite *Ensliniella parasitica* travels on female mason wasps *Allodynerus delphinalis*. Female wasps lay a single egg in a brood cell within a dead plant, and provision the cell with paralysed caterpillars and a few phoretic mites. The mites are mildly parasitic because they feed on the developing wasp's haemolymph (*Okabe and Shun'ichi, 2008*). However, if the wasp pupae are threatened by parasitoid wasps, the mite protects them from attack, thus switching from parasite to mutualist (*Okabe and Shun'ichi, 2008*). Nevertheless phoretic interactions are generally under-studied and their capacity to extend into further interactions that influence host fitness remains poorly understood (*White et al., 2017*).

For burying beetles, their phoretic relationship with *P. carabi* mites changes once the beetle has located the dead body. This study focuses entirely on the interactions that take place from that point onwards, during reproduction. The intimate association between beetles and mites continues through frequent contact as the two species breed alongside each other on the small dead body, and this enables each party to influence the other's fitness. We characterize the changing relationship between the mite and the beetle by measuring the fitness outcome for each of them (*Figure 1—figure supplement 1*).

The beetle has a net positive effect on mite fitness. Without the beetle, the mite would not be able to breed at all. Furthermore, mites have greater reproductive success on beetle-prepared carrion than on other dead meat (*Sun and Kilner, 2019*). However, in some contexts, the mite reduces burying beetle fitness. Mite offspring compete with burying beetle larvae for resources on the carcass, and can directly predate upon beetle eggs and newly-hatched larvae (*Wilson, 1983*; *Beninger, 1993*; *De Gasperin and Kilner, 2015*). Thus, in some contexts the mites are harmful for the burying beetle.

In other contexts, though, the mite can potentially become a protective mutualist by defending burying beetle reproductive success when it is threatened by an enemy species (*Wilson, 1983*). Blowflies (Calliphoridae) are a particular competitive threat for burying beetles (*Scott, 1994*; *Sun et al., 2014*) because they can locate the newly dead more rapidly than burying beetles (within a few hours: *Shelomi et al., 2012*); personal observations) and start to lay eggs within minutes of arriving on the dead body (*Bornemissza, 1957*; *Matuszewski et al., 2010*; *Payne, 1965*). Mites can potentially prevent burying beetles from losing fitness to rival blowflies by eating blowfly eggs (*Springett, 1968*). As an indirect effect of the mites' predatory actions, the net fitness outcome of

the mite-beetle interaction becomes positive-positive. Since the mite is only able to feed upon blowflies because it was transported to the carrion by the burying beetle, the mite becomes a mutualist.

Two other factors additionally seem likely to determine whether mites have negative or positive effects on the fitness of their burying beetle hosts: temperature and mite density per host. Previous work has shown that at higher temperatures blowflies pose a greater threat to burying beetle and mite fitness. Blowflies are more abundant on carrion at higher temperatures, develop more rapidly and have higher reproductive success (*Sun et al., 2014*; *Wall et al., 1992*). High densities of mites might be more effective at protecting from blowflies under these conditions (*Okabe and Shun'ichi, 2008*). Yet high densities of phoretic mites and phoretic nematodes are also known to reduce the number and quality of burying beetle larvae produced, potentially making mites more harmful (*De Gasperin and Kilner, 2016*; *Wang and Rozen, 2019*). Therefore it is unclear how these three factors (temperature, mite density, and the presence of blowflies) interact to determine whether interactions between mites and their burying beetles are harmful to beetles or more mutualistic.

We used field and laboratory experiments on burying beetles and their *P. carabi* mites to determine how the effects of blowflies, temperature and mite density combine to influence the expression of a protective mutualism. Our experiments were designed specifically to investigate whether: 1) the presence of blowflies causes mites to switch from being harmful to becoming protective mutualists; 2) whether any transition to and from mutualism is modulated by temperature; and 3) whether any transition is additionally mediated by the density of mites on the carrion.

## Results

### Complementary patterns of reproductive success in burying beetles and blowflies, in the field

We found that the reproductive success of burying beetles and blowflies varied with temperature, though in a complementary pattern (*Figure 1A and B*). Whereas burying beetle reproductive success peaked at intermediate temperatures, and dipped at lower and higher temperatures (*Figure 1A* and *Supplementary file 1a*), blowflies had greatest reproductive success at lower and higher temperatures and much less success at intermediate temperatures (*Figure 1B* and *Supplementary file 1a*).

### Mites enhance burying beetle fitness in the field when there are blowflies present, but the effect depends on temperature and mite density

Adding mites to the breeding event changed these relationships, for both beetles and blowflies, though in different ways at different mite densities. When we added 10 mites, there was little effect on the overall reproductive success of beetles (*Figure 1C*; *Supplementary file 1a*), though mites significantly reduced the reproductive success of the blowflies at lower and higher temperature ranges (*Figure 1D*; *Supplementary file 1a*). When we added 20 mites, however, mites were especially effective at promoting beetle reproductive success at these same lower and higher temperatures (*Figure 1E*; *Supplementary file 1a*). Once again, they caused a corresponding decline in the success of blowflies breeding at lower and higher temperatures (*Figure 1F*; *Supplementary file 1a*).

Turning to the mites' perspective, we found that variation in their reproductive success could not be explained by temperature (*Supplementary file 1a*). From these initial results, we conclude that mites act as protective mutualists for burying beetles against blowflies in natural breeding conditions, matching results obtained previously for a different burying beetle species (*Wilson, 1983*), and that their effects are contingent on mite density per breeding event. Our results extend the findings of previous work by showing that mites promote burying beetle reproductive success specifically at lower and higher temperatures.

### Complementary patterns of reproductive success in burying beetles and blowflies are induced by each other in the lab

Next, we analysed data from Laboratory Experiment 1, focusing first on the effects of blowflies on burying beetle reproductive success, when there were no mites present (*Figure 2A* v. 2D). We found that blowflies reduced burying beetle reproductive success at lower and higher temperatures

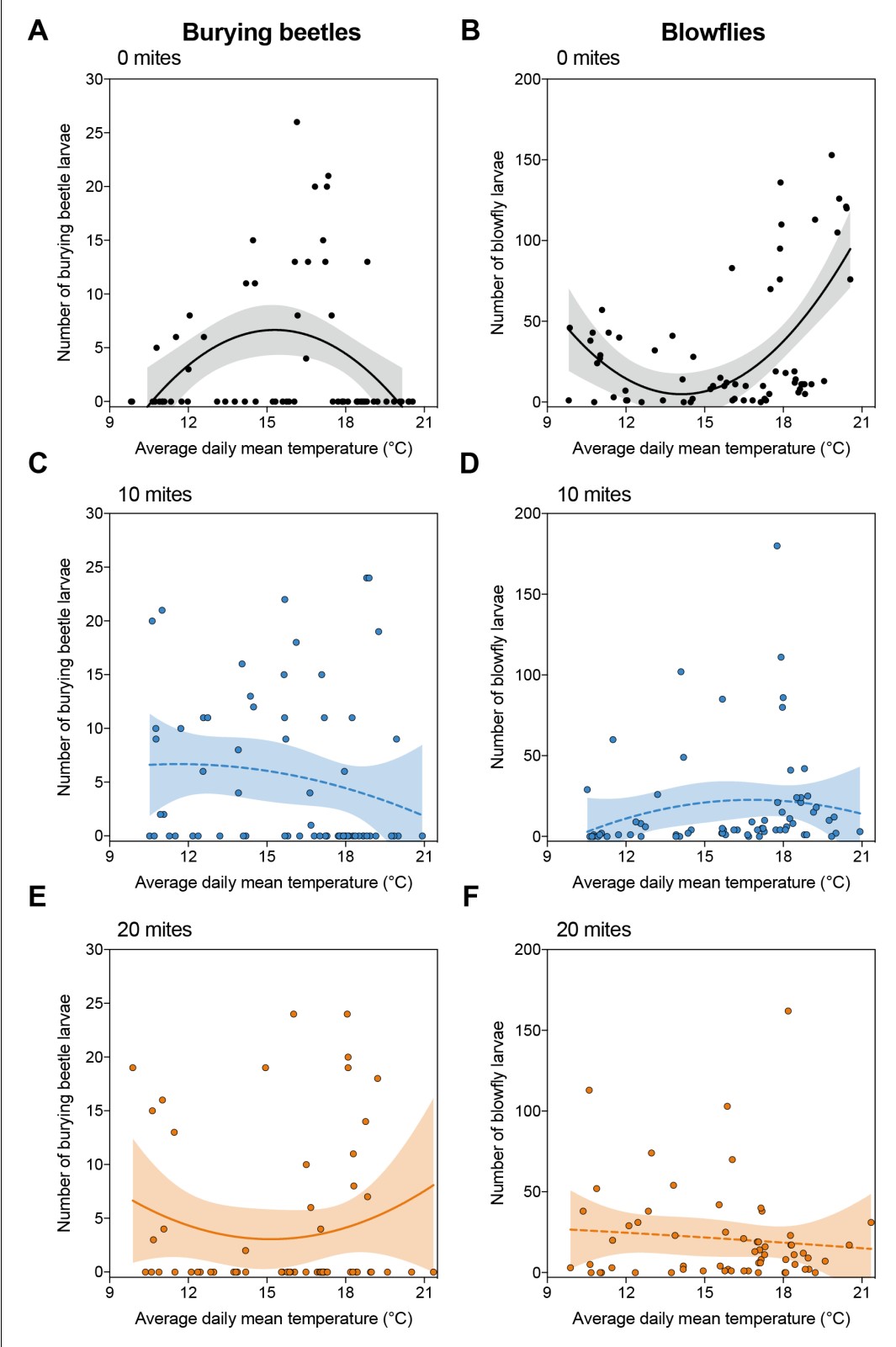

**Figure 1.** Reproductive success of burying beetles and blowflies under field conditions in relation to ambient air temperature, across the three different mite treatments. Shaded regions represent 95% confidence intervals, and solid and dashed lines represent statistically significant and non-significant regression lines from GLMM, respectively. Each datapoint represents one breeding event.

*Figure 1 continued on next page*

*Figure 1 continued*

The online version of this article includes the following figure supplement(s) for figure 1:

**Figure supplement 1.** Spatial distribution of breeding sites (yellow dots) used in the field experiment at the study in Madingley Wood, Cambridge, UK (Latitude: 52.22730°; Longitude: 0.04442°).

**Figure supplement 2.** Schematic side-view representation of the experimental setup used for each breeding event in the field (dimensions are in cm).

(interaction blowfly treatment x temperature treatment, $\chi^2$ = 25.85, d.f. = 2, p<0.001), and that blowflies caused greater reduction at higher temperatures than at lower temperatures (*post-hoc* comparison high v. low, $z = -2.47$, p=0.036).

To determine whether beetles likewise influenced blowfly reproductive success, we compared the number of blowfly larvae produced in Laboratory Experiment 1 with the number of blowfly larvae produced in Laboratory Experiment 2, when there were no beetles present. We found that burying beetles substantially reduced blowfly reproductive success but that the effect was temperature-dependent (interaction beetle x temperature treatments: $\chi^2$ = 38.32, d.f. = 2, p<0.001). Blowfly reproductive success was most strongly reduced by beetles at intermediate temperatures ($z = 10.59$, p<0.001), with a less pronounced decrease at lower temperatures ($z = 9.40$, p<0.001), and the least change of all at higher temperatures ($z = 7.04$, p<0.001).

## Blowflies are enemies to mites

Further analyses of Laboratory Experiment 1 revealed that blowflies reduced mite reproductive success (*Figure 2—figure supplement 2*; *Supplementary file 1b*) and that the extent of mite fitness loss was modulated by temperature (*Supplementary file 1b*). We found that blowflies reduced mite reproductive success at mid and higher temperatures (mid temperatures: *post-hoc* comparison without blowflies v. with blowflies, $z = 2.24$, p=0.025; higher temperatures: *post-hoc* comparison without blowflies v. with blowflies, $z = 3.29$, p=0.001). However, blowflies had no effect on mite reproductive success at lower temperatures (*post-hoc* comparison without blowflies v. with blowflies, $z = 0.30$, p=0.766). Temperature thus modulates the negative effects of the blowfly on both burying beetle and mite fitness (*Supplementary file 1b*).

## In the lab, mites reduce burying beetle fitness at high densities when blowflies are absent

Adding mites generally reduced burying beetle reproductive success, though to different degrees at different mite densities (*Figure 2A-C*; *Supplementary file 1b*). Across all temperatures, mites had no effect on beetle reproductive success in groups of 10 (*post-hoc* comparison 0 v. 10 mites, $z = 1.49$, p=0.298). However, adding 20 mites significantly reduced beetle reproductive success (*post-hoc* comparison 0 v. 20 mites, $z = 3.20$, p=0.004). Therefore, mites have mildly negative effects on burying beetle fitness, as has been reported before in previous work on *N. vespilloides* (*Beninger, 1993*; *De Gasperin and Kilner, 2015*; *Nehring et al., 2017*; *Sun et al., 2019*) and other *Nicrophorus* species (*Wilson and Knollenberg, 1987*).

Nevertheless, the loss in beetle reproductive success caused by mites at high temperatures was much less than that induced by blowflies (*post-hoc* comparison 0 mites, with blowflies v. 10 mites, without blowflies, $z = -3.61$, p=0.002; *post-hoc* comparison 0 mites, with blowflies v. 20 mites, without blowflies, $z = -2.85$, p=0.023).

## Mites switch from being harmful to mutualistic at lower and higher temperatures

We found that the presence of blowflies caused mites to switch to becoming more mutualistic. Furthermore, the extent of mutualism was dependent both on temperature and mite density, matching our findings in the field. At lower temperatures, neither density of mites affected beetle reproductive success when blowflies were present (*post-hoc* comparison 0 v. 10 mites, $z = -0.77$, p=0.720; *Figure 2E*; *post-hoc* comparison 0 v. 20 mites, $z = -0.60$, p=0.822; *Figure 2F*). At higher temperatures, 10 mites had no effect on burying beetle reproductive success either (*post-hoc* comparison 0 v. 10 mites, $z = -1.03$, p=0.560; *Figure 2E*). However, when 20 mites were added to the breeding

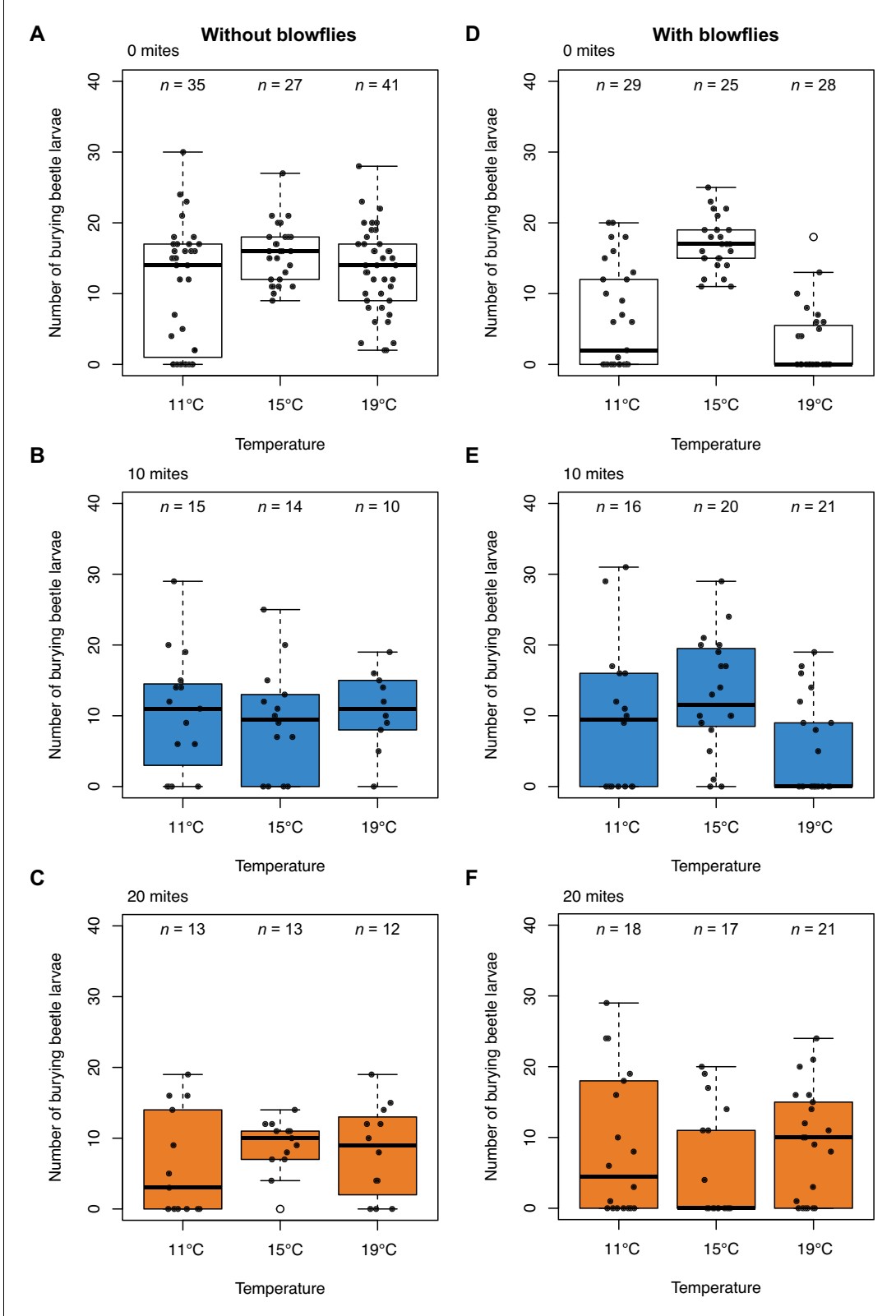

**Figure 2.** Burying beetle reproductive success under lab conditions in relation to ambient air temperature in the incubator, without and with blowflies, and across three different mite treatments. Sample sizes are shown above each boxplot. Boxplots show median (solid line), first quartile (bottom of box), third quartile (top of box), values that fall within 1.5 times of the interquartile range (dotted lines), and outliers (open circles). Each datapoint represents one breeding event.

*Figure 2 continued on next page*

*Figure 2 continued*

The online version of this article includes the following figure supplement(s) for figure 2:

**Figure supplement 1.** The daily mean, maximum, and minimum ambient air temperature in Madingley Woods during the field experiments conducted in 2016 and 2017.

**Figure supplement 2.** Reproductive success of mites in relation to temperature, without and with blowflies and across the temperature treatments.

event, they increased beetle reproductive success but only at higher temperatures (*post-hoc* comparison 0 v. 20 mites, $z = -3.04$, p=0.007; *Figure 2F*).

The increase in beetle reproductive success was matched by a corresponding mite-induced decline in blowfly reproductive success (*Figure 3*), with the pattern of decline again matching the results of our field experiment (*Figure 1B*). When there were no mites present, blowflies breeding alongside burying beetles had much greater reproductive success at higher temperatures and lower temperatures than at intermediate temperatures (*post-hoc* comparison high v. mid temperature, $z = 5.61$, p<0.001; *post-hoc* comparison low v. mid temperature, $z = 3.21$, p=0.004; *Figure 3A*).

In summary, the field and lab experimental results each suggest that burying beetles can manage singlehandedly to defend their reproductive success against blowflies at intermediate temperatures, but that they struggle to produce as many larvae at higher and lower temperatures (*Figure 1B*, *Figure 2D*). These are the temperatures at which blowflies have highest reproductive success when there are no mites present. Although adding 10 mites did not cause a significant reduction in the number of blowfly larvae produced (lower temperatures: *post-hoc* comparison 0 v. 10 mites, $z = 1.76$, p=0.183; higher temperatures: *post-hoc* comparison 0 v. 10 mites, $z = -0.65$, p=0.792; *Figure 3B*), adding 20 mites to the breeding event caused blowflies to perform badly at all temperatures (*Figure 3C*).

How are burying beetles (at intermediate temperatures) and mites (at lower and higher temperatures) able to cause such a reduction in blowfly reproductive success? Both species wander all over the carrion nest, especially during carcass preparation before the burying beetle larvae hatch (*Smiseth et al., 2003*). They graze on the surface of the carrion as they go, and have been observed to consume blowflies when they are eggs or newly hatched 1st instar blowfly larvae (*Wilson, 1983*; *Wilson and Knollenberg, 1987*). The likelihood that blowfly eggs will be eaten therefore depends partly on the duration of these vulnerable early life stages during blowfly development, and partly on the extent to which beetles and mites prey upon blowflies. We tested whether each is temperature dependent.

## At higher temperatures, blowflies evade attack through more rapid development

We found that temperature could not explain any variation in either blowfly reproductive success (*Figure 4—figure supplement 1*; *Supplementary file 1c*), or the extent to which blowfly larvae consumed the carcass (*Figure 4—figure supplement 1*; *Supplementary file 1c*). However, blowfly development was greatly accelerated at higher temperatures (*Figure 4A*; *Supplementary file 1c*), with blowflies spending significantly less time as eggs and 1st instar larvae at higher temperatures than at lower temperatures (eggs: $t = -3.76$, p<0.001; 1st: $t = -4.89$, p<0.001).

## At lower temperatures, beetle defences against blowflies are weaker

When we compared the number of blowfly larvae produced in Laboratory Experiment 2, when beetles were able to prepare a carcass, and Laboratory Experiment 3, when beetles were absent, we found that carcass preparation by beetles reduced the number of blowfly larvae produced and but that its effectiveness was sensitive to temperature (interaction carcass preparation x temperature treatments: $\chi^2 = 19.67$, d.f. = 2, p<0.001). Blowflies showed the greatest loss in fitness at intermediate temperatures ($z = 9.84$, p<0.001) with a less marked reduction in fitness at lower ($z = 5.16$, p<0.001) and higher temperatures ($z = 6.25$, p<0.001).

We found that the effectiveness of carcass preparation by beetles varied with temperature (*Figure 4B*; *Supplementary file 1d*). Specifically, beetles converted a dead body into a rounder nest for their larvae at both higher and mid temperatures than at lower temperatures (*post-hoc* comparison high v. low, $z = 4.68$, p<0.001; *post-hoc* comparison low v. mid, $z = -4.83$, p<0.001). The

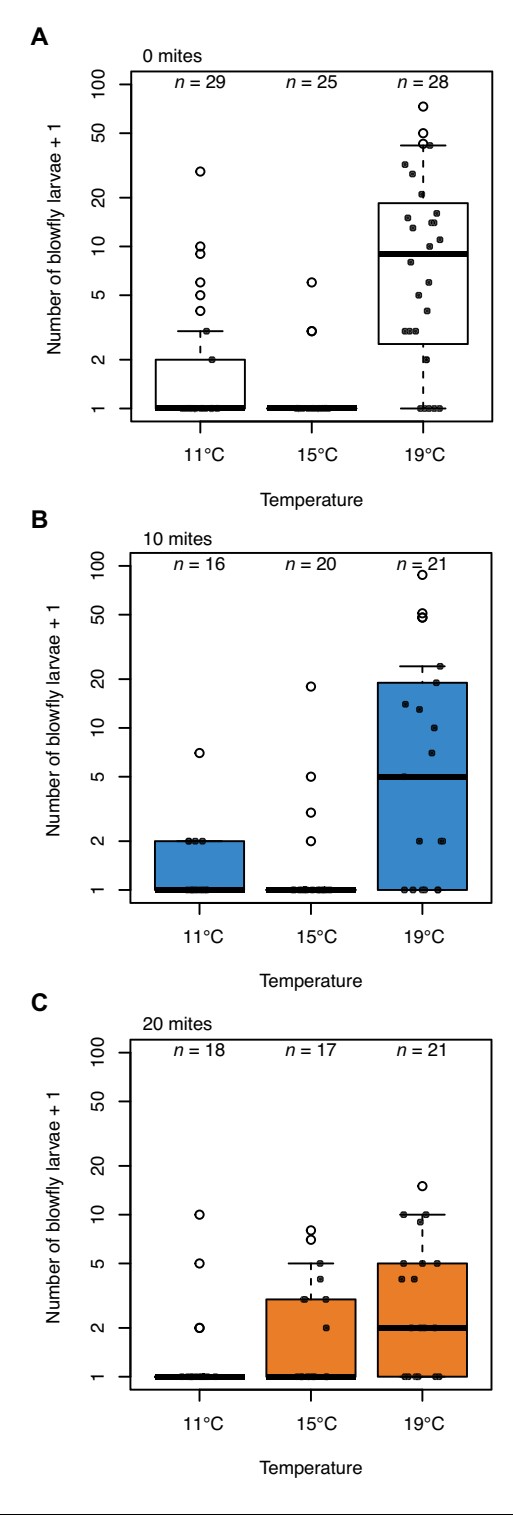

**Figure 3.** Blowfly reproductive success in relation to temperature in the presence of (A) 0 mites, (B) 10 mites and (C) 20 mites. Sample sizes are as indicated above each bar. Boxplots show median (solid line), first quartile (bottom of box), third quartile (top of box), values that fall within 1.5 times of the interquartile

*Figure 3 continued on next page*

rounder the prepared carcass was, the fewer the blowfly larvae that survived ($\chi^2 = 13.78$, d.f. = 1, p<0.001; *Figure 4C*).

The combined effects of temperature on both carcass preparation by beetles and blowfly development, explain why blowflies are able to produce more larvae at higher and lower temperatures than at mid temperatures - and therefore why they pose more of a threat to burying beetle and mite fitness at these temperatures. Burying beetles can singlehandedly defend themselves against blowflies at intermediate temperatures through their activities during carcass preparation. At higher temperatures, blowflies develop sufficiently rapidly that they can evade these beetle defences. At lower temperatures, burying beetles are less able to defend themselves against blowflies during carcass preparation.

## Discussion

The aim of this study was to determine how biotic and abiotic factors combine to influence the context-dependent expression of a protective mutualism, using the changeable interactions between burying beetles and their mites as a model system. Our experiments reveal a web of direct and indirect ecological interactions between burying beetles, *P. carabi* mites and blowflies as they breed alongside each other on small carrion (see *Figure 5*). The web is partly constructed by the burying beetles themselves, because they alone transport mites to the carrion. However, the interaction between burying beetles and their *P. carabi* mites depends on whether blowflies are present too - because predation by mites on blowfly eggs then indirectly enhances burying beetle reproductive success. The extent of mutualism also varies with increasing temperature stress, and with increasing mite density. All three factors cause a corresponding change in the net fitness outcome for burying beetles and this determines whether the mite harms burying beetle fitness or is more mutualistic (*Figure 5*).

### (1) Do blowflies cause mites to switch from being harmful to becoming protective mutualists?

Consistent with previous work on other burying beetle species (*Wilson, 1983*), we found that mites were antagonistic to beetles at all temperatures in the absence of blowflies (*Figure 2*). A similar decrease in the extent of mutualism has been detected in other protective mutualisms when the third-party enemy species is absent or

*Figure 3 continued*

range (dotted lines), and outliers (open circles). Each datapoint represents one breeding event.

removed (*Hopkins et al., 2017*). Then, it is common for the host to reduce the rewards it offers its protective mutualist (*Palmer et al., 2015*; *Palmer et al., 2008*). It is unclear whether this happens in burying beetles too. However, the main service that beetles offer to mites is transport to carrion. This means that the beetles' payment to the mites would have to be modulated either in advance of their protection service, when mites are transported to carrion, or retrospectively, when the adult beetles fly off carrying the mites' offspring with them at the end of reproduction. Either way, since the prevalence of blowflies is likely to vary locally from one breeding attempt to the next, it is hard to see how beetles could accurately modulate the transport service they offer to mites in relation to the prevalence of blowflies. An alternative possibility is that some of the other mite species carried by burying beetles in nature (which we excluded from our experiments), or the

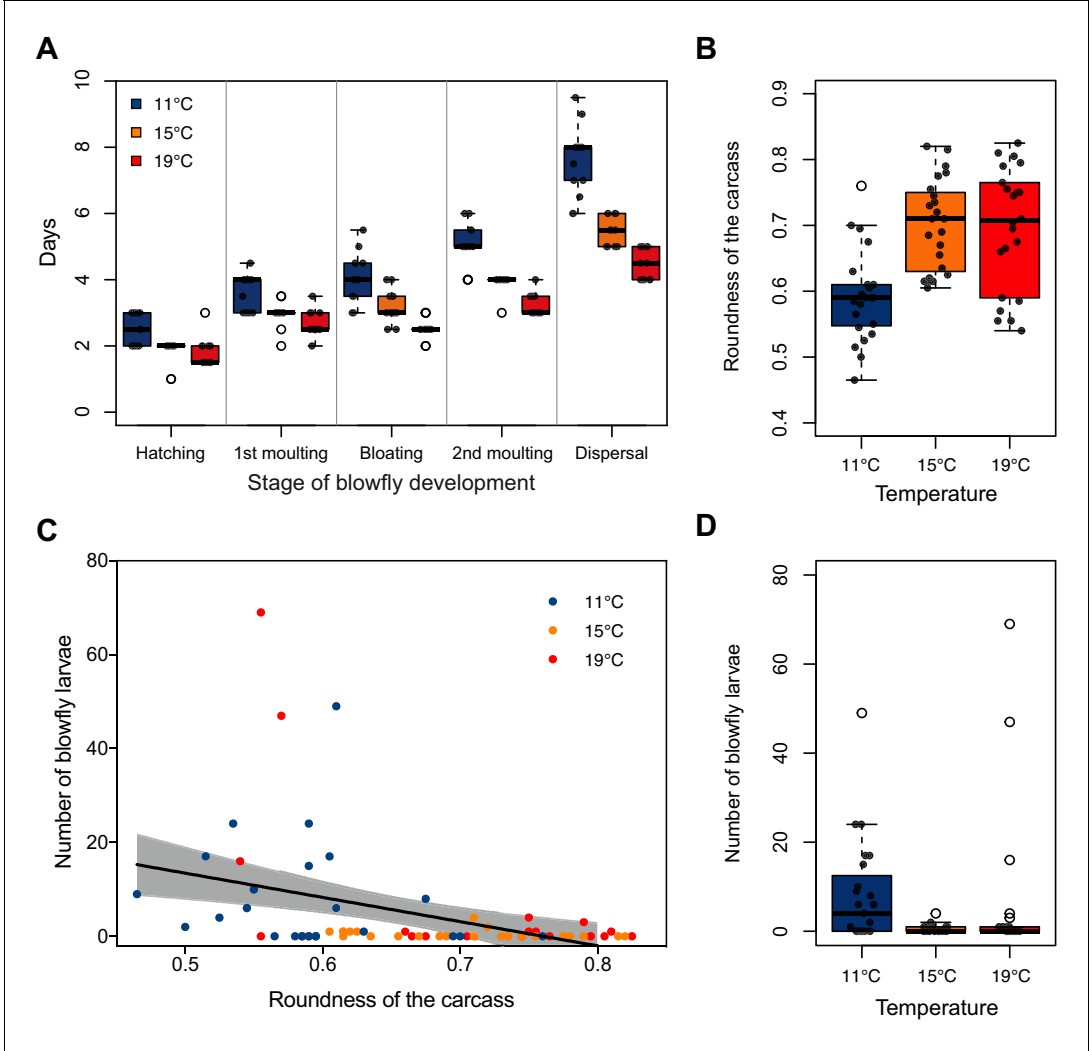

**Figure 4.** Effect of temperature on blowfly and burying beetle performance during carcass preparation. (**A**) The effect of temperature on blowfly development rate (*n* = 13 mouse carcasses for each temperature treatment) and (**B–D**) the relationship between number of blowfly larvae and roundness of the carcass for the low, mid, and high temperature treatment (*n* = 23, 23, and, 22 mouse carcasses, respectively). Boxplots show median (solid line), first quartile (bottom of box), third quartile (top of box), values that fall within 1.5 times of the interquartile range (dotted lines), and outliers (open circles). The shaded region represents 95% confidence interval, and the line represents statistically significant regression line from GLM. The online version of this article includes the following figure supplement(s) for figure 4:

**Figure supplement 1.** Effect of temperature on blowfly reproductive performance.

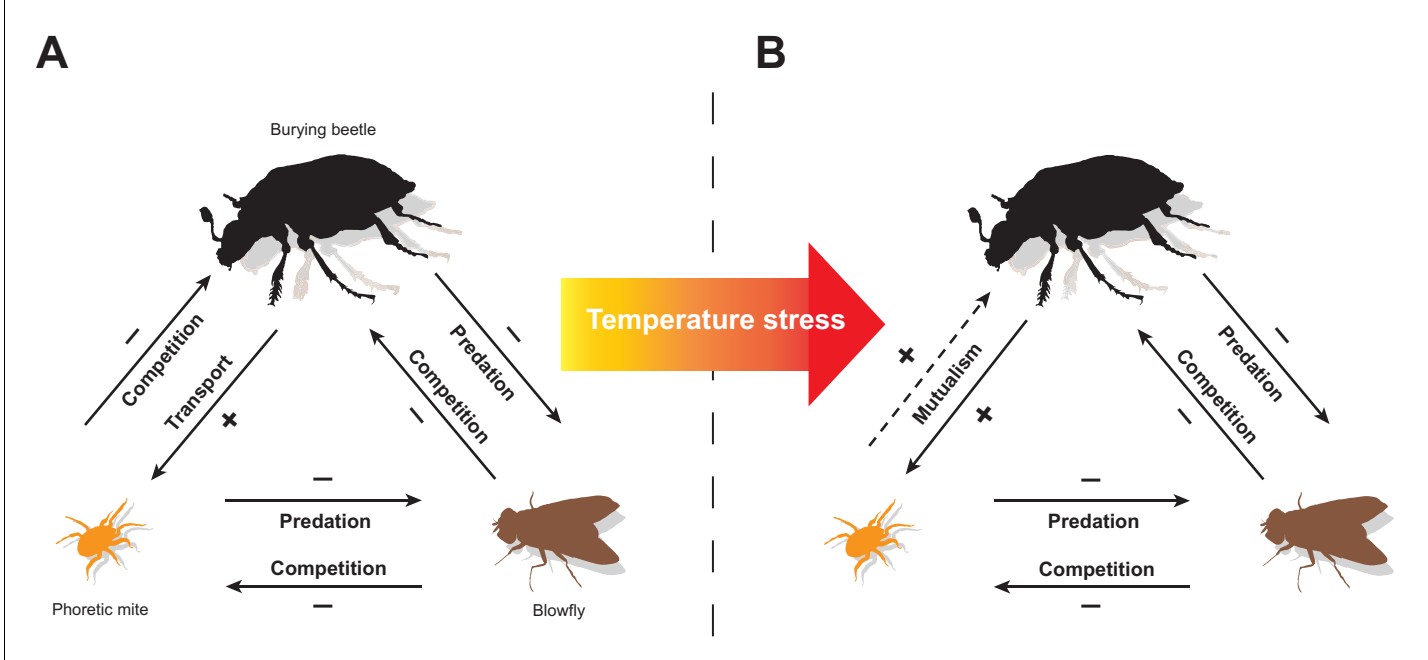

**Figure 5.** A summary of the experimental results, showing how the interactions between burying beetles, mites, and blowflies change in response to an increase in temperature stress (caused by temperatures that are higher or lower than average). Direct interactions between species are shown with solid lines while indirect interactions are shown with dashed lines. The arrow points to the species whose fitness is affected by the focal species. The signs (+/-) indicate positive or negative effects on fitness. Our overall conclusion is that a temperature-enhanced threat from blowflies causes mites to become protective mutualists of their burying beetle hosts.

phoretic nematodes that are also present upon the beetle (*Wang and Rozen, 2019*) modulate the harm inflicted by *P. carabi* on its burying beetle host. Whether this actually happens, however, remains to be determined in future work.

## (2) Is the expression of the protective mutualism modulated by temperature?

Previous studies have emphasised the significance of the abiotic environment in shifting the outcome of species interactions (*Chamberlain et al., 2014*; *Gorter et al., 2016*; *Hoeksema and Bruna, 2015*; *Hopkins et al., 2017*). Protective mutualisms sometimes break down at higher temperatures because the protecting partner is more vulnerable to heat stress when temperatures rise (*Barton and Ives, 2014*; *Doremus and Oliver, 2017*; *Fitzpatrick et al., 2014*). However, we found no evidence that mites were more vulnerable to higher temperatures, whether in field or laboratory conditions. Instead, the main driver of change in the protective mutualism came from the response of enemy blowflies, and the behaviour of the burying beetles themselves, to variation in temperature (*Figure 4*). We suggest that similar effects might be found in other protective mutualisms where enemy species are more likely to thrive at high temperatures, providing that both partners can tolerate some thermal stress. Predicting how populations might respond to more variable temperatures thus involves understanding its interactions within the natural ecological community as well as some knowledge of the intrinsic variation in the thermal tolerance of the mutualistic partner (*Early and Keith, 2019*).

## (3) Is the expression of the protective mutualism modulated by the density of mites?

The mites' capacity to defend burying beetles against competition from blowflies was both temperature-dependent and density-dependent. In the field and in the lab, blowflies posed a greater threat to burying beetle fitness at higher temperatures and then it took a high density of mites to neutralize this danger. Increased mite density has been found to influence the effectiveness of defences against

enemy species in other protective mutualisms as well (e.g. *Okabe and Shun'ichi, 2008*). Our experiments captured the likely variation in mite density at natural breeding events. However, we have no evidence to suggest that beetles can regulate the density of mites they carry in anticipation of the threats they face to their reproductive success (*Sun et al., 2019*).

In conclusion, we have shown how the expression of a protective mutualism between burying beetles and their *P. carabi* mites is context-dependent and depends on a complex interplay of biotic and abiotic factors. In common with other facultatively expressed mutualisms (*Afkhami et al., 2014*; *Johnson, 2015*; *Peay, 2016*), short-term variation in the expression of this protective mutualism may influence the capacity of its host burying beetle to persist in adverse environments.

## Materials and methods

### Burying beetles and phoretic mites in Madingley Wood

Fieldwork was carried out at Madingley Woods in Cambridgeshire UK, an ancient woodland (*Goldberg et al., 2007*) of mixed deciduous trees near the Sub-Department of Animal Behaviour, University of Cambridge, (Latitude: 52.22730°; Longitude: 0.04442°). We trapped *N. vespilloides* carrying the mite *P. carabi* by setting Japanese beetle traps, baited with ~30 g fresh mice, from June to October, 2016–2017. Ambient air temperature was recorded locally at 1 hr intervals using an iButton temperature data logger ($n$ = 8; DS1922L-F5#, Maxim Integrated Products, Inc), which was suspended alongside each trap at 1 m above the ground, and shielded from direct exposure to sunlight. Traps were checked daily to determine when the beetles first located the dead body within. The mean ± S.E.M. time to discovery was 3.42 ± 0.77 days. Each trap was emptied every two weeks, and re-baited with a fresh mouse carcass. At this point, we took the contents back to the lab and counted the total number of *N. vespilloides* caught in the trap and the number of *P. carabi* carried by each individual beetle. Beetles were temporarily anaesthetized using $CO_2$ and mites were then detached with a fine brush and tweezers. Field-caught burying beetles naturally carried a mean ± S.E.M. of 10.82 ± 0.45 mites (see *Figure 2—figure supplement 1* from *Sun et al., 2019* for frequency distribution of mite density), while 70% of them carried 1–20 mites ($n$ = 1369 beetles). Field-caught beetles, mites, and blowfly larvae collected from the traps were used to establish laboratory colonies (see below).

### Field experiment: how does burying beetle reproductive success covary with blowflies, mite density and ambient air temperature?

Experimental breeding events were staged in Madingley Woods. Breeding events were established at 20 different sites (see *Figure 1—figure supplement 1*), separated from each other by approx. 30 m. Each site was used more than once during the course of the burying beetle's breeding season. We recorded ambient temperature during each experiment by using iButton temperature data loggers placed at 1 m above ground at 1 hr intervals throughout. The set-up for each breeding event is shown in *Figure 1—figure supplement 2*. A 8–16 g (12.40 ± 0.15 g) mouse carcass was placed on the compost and left for three days, to simulate the average time taken by beetles to locate a carcass in the field (see above). Blowflies that were naturally present in the woodland were able to lay their eggs opportunistically on the mouse corpse too, while it remained above ground. We then added a pair of burying beetles from the laboratory colony. We also added mites from the lab colony at one of three different densities: 0 ($n$ = 66), 10 ($n$ = 68), or 20 mite ($n$ = 61) deutonymphs. We staged 195 breeding events in all. Each experiment was terminated either when the beetle larvae dispersed or when the dead body was completely consumed by blowfly larvae. At this point we measured components of beetle fitness (number of beetle larvae; see below), blowfly fitness (number of blowfly larvae), and mite fitness (number of dispersing mite deutonymphs on adult beetles).

### Maintenance of laboratory colonies of beetles, mites, blowflies
#### Burying beetles

We bred burying beetles by introducing pairs of unrelated males and females to a mouse carcass (7–15 g) in a plastic container (17 × 12 × 6 cm filled with 2 cm of moist soil). All larvae were counted and collected at dispersal, and transferred to eclosion boxes (10 × 10 × 2 cm, 25 compartments) filled with damp soil. Once they had developed into adults, beetles were kept individually in plastic

containers (12 × 8 × 2 cm) filled with moist soil, and were fed twice a week with small pieces of minced beef.

## Mites

We maintained mite colonies in plastic containers (17 × 12 × 6 cm filled with 2 cm of moist soil). Each container was provided with an adult beetle and fed with pieces of minced beef twice a week. We bred mites once a month by introducing 15 mite deutonymphs to a pair of beetles and a mouse carcass in plastic containers (17 × 12 × 6 cm filled with 2 cm of moist soil; $n$ = 10). When the burying beetle larvae had completed their development, we collected mite deutonymphs that were dispersing on adult beetles. Newly-emerged mites were reintroduced to the containers holding the mite colony.

## Blowflies

Colonies of blowflies *Calliphora vomitoria* ($n$ = 5 colonies) were reared in screened cages (32.5 × 32.5 × 32.5 cm). They were continuously supplied with a mixture of powdered milk and dry granulated sugar, and ad lib. water. We fed newly emerged blowflies with pig liver to induce maturation of the flies' ovaries. After a week, these blowflies were then given mouse carcasses to breed upon. All beetle, mite, and blowfly colonies were kept at 21 ± 2°C with a photoperiod of 16:8 light:dark.

## Laboratory Experiment 1: manipulations of blowflies, mites and temperature

To understand how temperature and mite density together mediate blowfly competition with burying beetles, we repeated the field experiment in a lab setting so that we could manipulate temperature and the presence of blowflies as well as mite density.

### Manipulating the presence/absence of blowflies

We placed 30 mg (30.22 ± 0.07 mg) newly-laid blowfly eggs onto a 7–16 g (11.13 ± 0.15 g) mouse carcass before giving it to beetles to breed upon, to mimic the rapid oviposition by blowflies in nature on a freshly dead vertebrate (*Wilson, 1983*). As a control, dead mice of similar size (10.64 ± 0.15 g) were kept free of blowflies. In both blowfly treatments, the dead mouse was placed on the soil in a breeding box in a temperature-regulated breeding chamber for 3 days before adding the beetles, simulating the later arrival time of the beetle at the carcass that is seen in nature (see above). During this time, the fly eggs were able to hatch and the blowfly larvae started to consume the carcass.

### Manipulations of mite density

We used the same treatment as in the field experiment: 0, 10, or 20 mites. Mite deutonymphs were introduced to the dead mouse at the same time as the burying beetles.

### Manipulations of temperature

The six treatments described above were each staged in temperature-regulated breeding chambers (Panasonic MLR-352-PE). Each temperature treatment mimicked the 8°C diurnal temperature fluctuation that is typical for Madingley Woods, during the burying beetle's breeding season (*Figure 2— figure supplement 1*). The mean temperature for each manipulation was 11, 15, and 19°C, which matches the mean seasonal low, intermediate, and high temperatures, respectively, in Madingley Woods (*Figure 2—figure supplement 1*). Each of the six treatments was carried at these three temperatures, generating a fully factorial experiment with 18 treatments in all (three mite treatments (0, 10 or 20 mites) x two blowfly treatments (blowfly or no blowfly) x three temperature treatments (11, 15, and 19°C). At the end of each breeding bout, indicated by either the beetle larvae starting to disperse away or carcass consumption by blowfly larvae, whichever came sooner, we measured the fitness components of beetles, mites, and blowflies using the methods described above in the field experiments. For logistical reasons, replicates of all 18 treatments were evenly spread over four blocks, carried out in succession.

## Laboratory Experiment 2: effect of temperature on blowfly development

To examine how blowflies respond to temperature, in the absence of the mites and the burying beetles, we counted the number of dispersing blowfly larvae, and the rate of carcass consumption, at the three different temperatures used in laboratory experiment 1 (11, 15, and 19°C; $n$ = 13 carcasses for each temperature treatment). Once again, we placed blowfly eggs (30.22 ± 0.09 mg) on a mouse carcass (10.74 ± 0.30 g) that sat on soil in a plastic breeding box, and put the box in a temperature-controlled breeding chamber. (No burying beetles or mites were added this time). Every 12 hr we checked the boxes and determined the stage of blowfly larval development attained, namely 1st, 2nd, 3rd instars and post-feeding. In addition, we recorded when the carcass entered the bloating stage (indicated by swelling and putrefaction). When the larvae entered the post-feeding stage, we counted them, and recorded their total mass. From these data, we determined the proportion of carcass consumed, calculated as total mass of larvae divided by initial carcass mass.

## Laboratory Experiment 3: effect of temperature on beetle defences against blowflies during carcass preparation

To understand the effect of temperature on the effectiveness of carcass preparation by burying beetles in defending against infestation by blowflies, we placed blowfly eggs (30.05 ± 0.09 mg) on a mouse carcass (13.25 ± 0.24 g) prior to introducing pairs of beetles at three different temperatures (11, 15, and 19°C; $n$ = 23, 23, 22 carcasses for each temperature treatment, respectively). This time, each carcass was transferred to a new plastic breeding box once the beetles had completed carcass preparation but before their eggs had hatched. Once the carcass had been moved, it was kept at the same intermediate temperature regardless of the temperature treatment previously experienced during carcass preparation. This allowed us to isolate the effects of temperature on beetle carcass preparation, and its relation to subsequent blowfly fitness.

We quantified the extent of carcass preparation by measuring the sphericity of each prepared carcass, using previously established methods (*De Gasperin et al., 2016*), calculating roundness from a two-dimensional proxy. Each carcass was photographed against a white background from the top and the side using two identical digital cameras (Fuji lm av200), each kept at a constant distance of 30 cm to the carcass. We processed the images with white circle to remove legs, tails, and large pieces of soil in GIMP (version 2.6.11), prior to roundness analysis. We estimated the roundness from each image using a boundary tracing routine, *bwboundaries*, in Matlab (The Mathworks, USA). Each image was separated from the white background with a filter of 5 pixels to remove the smallest details, such as hairs and soils smaller than 1 mm (the photographs taken from the top and side were 6.4 and 6.36 pixels per mm, respectively). The roundness was then determined by calculating a metric, $4\pi*area/perimeter^2$, in which a score of 1 denotes a perfect circle. An overall roundness score was derived by averaging roundness of the top and the side images of each carcass.

## Statistical analyses

Generalised linear mixed model (GLMM) analyses were carried out in the statistical programme R 3.4.3 using the package *lme4* (*Bates et al., 2015*). Model formulae are given in the tables of results (see *Supplementary file 1*). Non-significant interaction terms were dropped from the analyses before deriving the final model. As is common statistical practice (e.g. *Gelman and Hill, 2007*), if we found a significant interaction term, we split the dataset accordingly to determine how the interaction arose. Power analyses were performed based on 1000 Monte Carlo simulations, with the function powerSim in the package *SIMR* (*Green and MacLeod, 2016*).

## Field experiment

We sought correlates of beetle brood size, the number of blowfly larvae, and the number of mite offspring number at the end of each trial, using separate GLMMs each with negative binomial distributions. For the models with beetle brood size and the number of blowfly larvae as independent variables, we included the variables carcass mass, mite treatment (0, 10, 20 mites), temperature, and the interaction between mite treatment and temperature. Mite treatment was a categorical factor, whereas carcass mass and temperature were continuous variables. Temperature was calculated as the average daily mean temperature, from carcass presentation to larval dispersal (or carcass

consumption by blowfly larvae). We also included a squared measure of temperature in the model because we found that the non-linear effects of temperature explained more variation than any linear effects. (We compared the performance of different models using the Akaike Information Criterion (AIC), using the function model.sel in the package *MuMIn,* and obtained the following results. Models of burying beetle reproductive success: with temperature as a non-linear variable: AICc = 802.2, Akaike weight = 0.93 v. with temperature as a linear variable: AICc = 807.4, Akaike weight = 0.07. Models of blowfly reproductive success: with temperature as a non-linear variable: AICc = 1541, Akaike weight = 0.99 v. with temperature as a linear variable: AICc = 1550.2, Akaike weight = 0.01).

The model analysing mite reproductive success included data from the treatments with 10 and 20 mites and included carcass mass and temperature as covariates. In all three models, experimental site and year were included as random factors.

## Laboratory experiments

We analysed the reproductive success of beetles, blowflies, and mites using GLMMs with a negative binomial distribution to account for data overdispersion. We also included block as a random factor. Post-hoc pairwise comparisons were performed using the package *lsmeans* (*Lenth, 2016*) if an interaction was detected; *p* value for post-hoc comparisons were adjusted using Tukey's honestly significant difference (HSD) method. The data from the field experiment revealed a non-linear relationship between temperature and measures of reproductive success (see *Figure 1*). Therefore, we conservatively analysed the effect of the three different temperature (11, 15, 19℃) by treating temperature as a categorical factor in all these models.

### Analyses of beetle reproductive success

We tested for the interacting effects of blowfly (yes/no), mite (0, 10, 20), and temperature (11, 15, 19℃) treatments on the reproductive success of beetles by including all three treatments as categorical factors. Separate GLMMs were used to make further comparisons between blowfly and mite treatments to determine how any significant interactions arose.

### Analyses of blowfly reproductive success

We tested for the interacting effects of mites (0, 10, 20) and temperature (11, 15, 19℃) treatments on the reproductive success of blowflies, and again by including them as categorical factors.

### Analyses of mite reproductive success

We tested for the interacting effects of blowfly (yes/no), mite (0, 10, 20) and temperature (11, 15, 19℃) treatments on the reproductive success of beetles. All three were included as categorical factors.

## Effect of temperature on blowfly larval development

We analysed the number of blowfly larvae in a negative binomial regression model with the function *glm.nb* in the MASS package to account for overdispersion. We analysed carcass consumption rate in a beta regression model in the *betareg* package. In both analyses, we included temperature treatment (11, 15, 19℃) as a categorical factor and blowfly egg mass and carcass mass as continuous variables. To analyse the effect of temperature on the developmental rate of blowfly larvae, we used a GLMM with Gaussian error structure and included the interaction between temperature treatment and developmental stage (both as categorical factors), blowfly egg mass, and carcass mass as continuous variables. In this analysis, we also included the ID of each carcass as a random factor, since carcasses were sampled repeatedly across different developmental stages.

## Effect of temperature on beetle's carcass preparation

We analysed the roundness of carcasses in a GLM and the number of blowfly larvae in a negative binomial regression model. In both analyses, temperature treatment (11, 15, 19℃) was included as a categorical factor, whereas blowfly egg mass and carcass mass were included as continuous variables. To further investigate the effects of carcass roundness on the number of blowfly larvae that

developed, we analysed the number of blowfly in a separate negative binomial regression model by additionally including roundness as a continuous variable.

## Acknowledgements

We are very grateful to the Editor and three anonymous referees for their constructive and insightful comments on earlier drafts of this paper. We would also like to thank E Turner, R Mashoodh and members of the Kilner Group for comments, and S Aspinall and C Swannack for substantial logistical support. This work was funded by a Rosemary Grant Award from the Society for the Study of Evolution. S-JS was supported by the Taiwan Cambridge Scholarship from the Cambridge Commonwealth, European and International Trust. RMK was supported by a European Research Council Consolidator grant 301785 BALDWINIAN_BEETLES and a Wolfson Merit Award from the Royal Society.

## Additional information

### Funding

| Funder | Grant reference number | Author |
| --- | --- | --- |
| Society for the Study of Evolution | Rosemary Grant Award | Syuan-Jyun Sun |
| Cambridge Commonwealth, European and International Trust | Taiwan Cambridge Scholarship | Syuan-Jyun Sun |
| European Research Council | Consolidator grant 301785 BALDWINIAN_BEETLES | Rebecca M Kilner |
| Royal Society | Wolfson Merit Award | Rebecca M Kilner |

The funders had no role in study design, data collection and interpretation, or the decision to submit the work for publication.

### Author contributions

Syuan-Jyun Sun, Conceptualization, Data curation, Formal analysis, Funding acquisition, Validation, Investigation, Visualization, Methodology, Writing - original draft, Project administration, Writing - review and editing; Rebecca M Kilner, Conceptualization, Resources, Supervision, Funding acquisition, Investigation, Methodology, Writing - original draft, Writing - review and editing

### Author ORCIDs

Syuan-Jyun Sun (iD) https://orcid.org/0000-0002-7859-9346
Rebecca M Kilner (iD) https://orcid.org/0000-0003-1159-0758

### Ethics

Animal experimentation: All of the animals were handled according to approved institutional animal care of the University of Cambridge. The protocol for field experimentation was approved by the Sub-Department of Animal Behaviour, University of Cambridge. During our experiments we handled our animals with care and they were not harmed at any stage. None of the animals that we used showed any signs of stress before, after or during the experiments.

### Decision letter and Author response

Decision letter https://doi.org/10.7554/eLife.55649.sa1
Author response https://doi.org/10.7554/eLife.55649.sa2

## Additional files

### Supplementary files

• Supplementary file 1. Results from the final models for each variable analysed. (**a**) Results from the final models for the reproductive success of beetles, blowflies, and mites in the field experiment. The final models used were: glmer.nb(Number of larvae ~ Mite treatment*(poly(temperature,degree = 2)[,2]+ poly(temperature,degree = 2)[,1])+Carcass mass+(1|site)+(1|year)). Models analyzing burying beetle larvae and blowfly larvae were both sufficient to reject the null hypotheses, with 81.3% and 98.6% power, respectively, whereas the model analyzing mite offspring was not, with a power of 36.9%. (**b**) Results from the final models for the reproductive success of beetles, blowflies, and mites in the Laboratory Experiment 1. For beetles, the final model used was: glmer.nb (Number of larvae ~ Mite treatment*Temperature treatment*Blowfly treatment+Carcass mass+(1| block)); for blowflies, the final model used was: glmer.nb(Number of larvae ~ Mite treatment*Temperature treatment+Carcass mass+(1|block)); and for mites, the final model used was: glmer.nb (Number of larvae ~ Blowfly treatment*Temperature treatment+Mite treatment+Carcass mass+(1| block)). All these models were sufficient to reject the null hypotheses, with the 97%, 97%, and 98.2% power, for analyses of burying beetle larvae, blowfly larvae, and mite offspring, respectively. (**c**) Results from the final models for the development of blowfly larvae in the Laboratory Experiment 2. For number of blowfly larvae, the final model used was: glm.nb(Number of larvae ~ Temperature treatment+Carcass mass+Blowfly egg mass); for carcass consumption rate, the final model used was: betareg(Consumption rate ~Temperature treatment+Carcass mass+Blowfly egg mass); and for development rate, the final model used was: glmer(Days ~ Temperature treatment*Developmental stage+Carcass mass+Blowfly egg mass+(1|carcass ID)). Models analyzing number of blowfly larvae and carcass consumption rate were both not sufficient to reject the null hypotheses, with 12.9% and 22.8% power, respectively, whereas the model analyzing development rate of blowfly larvae was highly sufficient, with a power of 100%. (**d**) Results from the final models for beetle's carcass preparation in the Laboratory Experiment 3. For number of blowfly larvae, the final model used was: glm. nb(Number of larvae ~ Temperature treatment+Carcass mass+Blowfly egg mass); and for carcass roundness, the final model used was: glm.nb(Roundness ~Temperature treatment+Carcass mass +Blowfly egg mass). Models analyzing number of blowfly larvae and carcass roundness were both sufficient to reject the null hypotheses, with 96.4% and 99.5% power, respectively.

• Transparent reporting form

### Data availability

All data generated or analysed during this study are included in the manuscript and supporting files. The data has also been deposited on Dryad (Sun, Syuan-Jyun; Kilner, Rebecca (2020), A temperature-enhanced threat from a common enemy converts a parasite into a mutualist, Dryad, Dataset, https://doi.org/10.5061/dryad.sj3tx961z).

The following dataset was generated:

| Author(s) | Year | Dataset title | Dataset URL | Database and Identifier |
|---|---|---|---|---|
| Syuan-Jyun S, Kilner RM | 2020 | Data from: Temperature stress induces mites to help their carrion beetle hosts by eliminating rival blowflies | https://doi.org/10.5061/dryad.sj3tx961z | Dryad Digital Repository, 10.5061/dryad.sj3tx961z |

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
