## [Decision Letter]

**Acceptance summary:**

This work on the interactions of burying beetles, mites and blowflies with temperature is a nice example of how multi-trophic relationships can change in response to an environmental factor and turn the sign of interactions around (from negative to positive). The combination of field work and experimental confirmation is strongly supporting the authors conclusions. Most interesting is the observation that at extreme temperatures the mites enhance beetle breeding success by eating blowfly eggs, thus turning the negative effect of mites in the absence of blowflies, into a positive effect.

**Decision letter after peer review:**

Thank you for submitting your article "A temperature-enhanced threat from a common enemy converts a parasite into a mutualist" for consideration by *eLife*. Your article has been reviewed by three peer reviewers, one of whom is a member of our Board of Reviewing Editors, and the evaluation has been overseen by Christian Rutz as the Senior Editor. The reviewers have opted to remain anonymous.

The reviewers have discussed their reviews with one another, and the Reviewing Editor has drafted this decision letter to help you prepare a revised submission.

This paper studies a complex trophic web between carrion beetles, phoretic mites and blowflies. The mites and beetles share carrion located by the beetles. The third carrion feeder, blowflies, compete with the other two carrion consumers. Both beetles and mites eat fly larvae as well as carrion. Flies are more common when it is warm and do better because they spend time in less vulnerable stages. They do better when it is cold because beetles are less able to roll the carcass into a ball to interfere with blowfly success. Beetles compete poorly with flies when it is too cold or too warm. Mites can eat blowfly eggs, thereby reducing the beetle's main competitor. The authors further consider if the net effect of mites on beetles is affected by flies and whether this net effect changes with temperature. It does, and the mechanism for this environmental interaction is mostly well explained.

The three reviewers have raised a variety of issues concerning the presentation of the study. Here is a short summary, and further details are provided in the three separate reports appended below.

1) Terminology. The general presentation of the work needs some revision. The reviewers made suggestions regarding the title and various terminological issues.

2) Framework. The placement of the work in a parasitism-mutualism framework was strongly criticised. The study would not lose any appeal if it was instead placed in a more conventional ecological framework but would gain in clarity.

3) Statistical issues. The Akaike Information Criterion is central for distinguishing between different statistical models. It is mentioned in the text but apparently not applied.

4) Literature cited. In connection with point 2 (above), it will be necessary to cover relevant literature when placing the work in a food-web theoretical framework.

These issues, including those listed below, should be taken care of in a revised version.

Reviewer #1:

This manuscript describes the breeding success of burying beetles in the presence and absence of two competing species, mites and blow flies. A combination of field and laboratory experiments shows that in the absence of blow flies, mites are harmful, while in the presence of blow flies, mites are-at extreme temperatures-beneficial to the beetles. The study is placed in a framework of parasitism – mutualism continuum, with the take home message that temperature can shift the form of the interaction from parasitism to mutualism.

I like the topic, the study and the way the data where produced. I have however problems with some of the approaches taken by the authors to analyse and present their data.

1) The title starts with "A changing climate…". This is far-fetched. The term climate shows only once up in the entire text (at the very end of the Discussion section). This title seems more aiming to catch attention than to give an adequate title for the study. I find this misleading and suggest changing this.

2) The parasitism – mutualism framework is a far fetch. The authors justify their claim by citing a dictionary entry (supplement) about the term parasitism. Even so this entry is ok, it is not how it is perceived in the general community and I think therefore it is misleading. I don't understand why it has to be parasitism. The message of the study remains the same if one states that species interactions change sign (from positive to negative) when the environmental conditions change. The main interaction is that mites compete with the beetles for food, which is competition. Adding the blow flies to the picture turns the mites into predators of blow fly eggs. This is helpful for the host. So, it is a change from a competitor to a mutualist.

3) Statistics: The authors show the models in the legends to the tables. But this is not complete. What are the different items in the model? Which of the items are factors, which are continuous variables? In the text some info is given, but is not very clear. What is a "categorical covariates". Is this a factor (like presence/absence of blow flies) or is this a continuous covariable (like temperature and carcass mass)? Please clarify. Temperature should be used throughout as a contiguous covariable, not as a factor. Furthermore, it would be helpful if the error distribution is given for each model. Given that the data have rather special error distributions (see figures) this is very important.

4) The authors state that they used "Akaike Information Criterion (AIC) by evaluating AICc, δ AICc and Akaike weights". I did not find any evidence in the statistics that this was actually done. This is important in particular for the fitting of the non-linear relationships.

Reviewer #2:

This study is very timely and unique, as it utilizes a combination of both field and lab work to examine protective mutualism using eukaryotes. A few comments:

1) "Transition", which is used throughout the manuscript can refer to evolutionary history/trajectory or an ecological-time, context-dependent switch. Given the combination of references used herein on evolutionary and ecological transitions, we strongly recommend clarifying throughout and choosing references/wording more carefully. There are numerous references to condition-dependent transitions along the parasite-mutualist continuum across the tree of life which are not mentioned.

2) There was also a lack of discussion of protective symbioses against abiotic stresses (like temperature) in the Introduction, which are different than the biotic stresses.

Here are some references on protective microbes against abiotic stresses:

Corbin.et al., 2017 - This reference was cited, but not sure the point of this paper was how it was used in the manuscript (–Introduction, "Yet theoretical analyses that consider how such interactions evolve and persist derive mainly from recent interest, with a particular focus on the microbial endosymbionts that can be induced to defend their hosts from attack"). This is a good source for examples of symbionts that protect hosts from temperature stresses and how temperature affects the association, not about biotic factors, which is what the sentence in the manuscript implies.

Engl et al., 2018.

Feldhaar, 2011.

Hoang, Gerardo and Morran, 2019.

3) Introduction is a bit sparse. For example, before the last paragraph, a brief sentence or two to summarize the big picture and tie to what is known to what is not known. Moreover, there was no mention of mutualist/symbiont density or mite density in the Introduction, even though that was what was driving the results

4) A diagram could be used to show how each player interacts with one another, especially because there are several definitions being used and different environmental contexts involved. For example, in the Introduction: being a mutualist is not mutually exclusive of being predatory. Mites are mutualistic towards the beetles and predatory towards the blowflies. A diagram can represent this. Moreover, doesn't the definition of parasitism have to be in a supplement (seems like an odd place) as opposed to just mentioned when the system is introduced?

5) Another point to mention in the Discussion section would be how the association becomes more mutualistic when there are more mites: the results indicate that more mites increase beetle fitness, which in turn benefit the mites (at the least, in order for beetle fitness to increase there has to be more mites. I'm not saying that the beetles directly promote mite fitness, but there is an association there. Also, perhaps more beetles could mean more mites can hitchhike?).

6) How common is it for phoresy to be parasitic? Worth mentioning how your consideration of the costs of mites squares with phoresy as currently understood in the literature.

7) Supplementary file 1, Supplementary file 2, Supplementary file 3 and Supplementary file 4, but particularly Supplementary file 3. Given the size of these models, do the authors have enough power to detect significant differences?

Reviewer #3:

This paper studies a complex trophic web between carrion beetles and phoretic mites. The mites and beetles share carrion located by the beetle, third carrion feeder, blowflies, competes with the other two carrion consumers. Both beetles and mites eat fly larvae as well as carrion. Flies are more common when it is warm and do better because they spend time in less vulnerable stages. They do better when it is cold because beetles are less able to roll the carcass into a ball to interfere with blowfly success. Beetles compete poorly with flies when it is too cold or too warm. Mites can eat blowfly eggs, thereby reducing the beetle's main competitor. The authors further consider if the net effect of mites on beetles is affected by flies and whether this net effect changes with temperature. It does, and the mechanism for this environmental interaction is well explained.

The paper is relatively well written and the authors are to be commended for their use of manipulative field and laboratory experiments. I enjoyed reading the paper and learned a lot. I do, however, have several suggestions that I think will improve the paper.

The title is vague. Authors sometimes assume that this will help the paper appear general and therefore capture broader readership. It doesn't. At least I would not bother to read it based on the title. And I think you want people like me to read it. A good title should state the key results and identify the system. I would read a paper titled something like: Phoretic mites help their carrion beetle hosts under temperature stress by eating competing blowfly larvae.

A general point about the Introduction: the paper will flow better (and read more like a scientific paper) if the authors clearly state the series of predictions they are going to test an how these predictions flow from their hypotheses. Each P-value in the result should be an evaluation of either an apriori prediction or the assessment of an assumption, or a controlling variable. And it should be clear how the hypotheses derive from the literature, theory or field observations.

The paper is missing some key information on what happens in nature. In particular, what is the natural distribution of mites on beetles in the field (this will help readers understand whether the treatments in the experiments are reasonable).

This gets me to my point about whether the mite is a parasite or a mutualist or a whatever. This might be semantics, but I put a high bar on assigning parasitism. And the beetle-mite interaction does not cross that bar for me. Namely, the negative trophic interaction does not appear to occur during intimacy. Parasitism must be defined at the stage level rather than the species level. I also put a particularly high bar for deciding that parasitism is context dependent when simpler explanations via indirect effects are possible. To me it is clear that the phoretic mite stage is not a parasite because it does not feed on the beetle and the free-living mite stage is not a parasite because it neither feeds on the beetle or is intimate with the beetle. This does not mean the interaction is not interesting. In fact, I think it is far more interesting than the way the authors have packaged it. The mite is a phoretic commensal that uses its host to be transported to a common food source. Once on that common food source, the relationship can become competitive. But that competition can lead to a net indirect benefit under situations where another competitor (blowflies) might dominate. This interpretation is better couched in food-web theory rather than evolutionary theory.

As an example, here is a paper I remembered from decades ago that pondered over the relationship between mites and hosts and third-party interactions. I include the citation for the authors' interest. This author had similar observations but did not conclude a plastic parasitic-mutualistic relationship.

Rigby, (1996).

[Editors' note: further revisions were suggested prior to acceptance, as described below.]

Thank you for submitting your article "Temperature stress induces mites to help their carrion beetle hosts by eliminating rival blowflies" for consideration by *eLife*. Your article has been reviewed again by a Senior Editor, a Reviewing Editor, and two reviewers.

Following a consultation discussion, the Reviewing Editor has drafted this decision letter to help you prepare a revised submission.

The editors have judged that your manuscript is of interest, but as described below, feel that additional revision is required before it can be published. We are also offering, if you choose, to post your manuscript to bioRxiv (if it is not already there) along with this decision letter and a formal designation that the manuscript is "in revision at *eLife*". Please let us know if you would like to pursue this option.

The reviewers were mostly satisfied with your responses to their comments from the first round of review. However, there are still points that require further attention. In particular, both reviewers remained unconvinced by your assertion that this is a host-parasite system. We suggest the following course of action for addressing this concern: We feel there is no need to stress the host-parasite framework in the abstract, introduction, and result section – the work is very interesting without this. If you wish, you can add a short paragraph to the Discussion section that outlines the reasons why you think this system can also be viewed productively within a host-parasite framework. In this section, if you choose to include it, please contrast your interpretation with alternative views, as presented by the reviewers – that way, readers can hear both sides of the argument.

Further comments from reviewers #1 and #3 follow.

Reviewer #1:

The revised manuscript fixed many of the issues I had with the earlier version. The authors did a nice job. There are still two points I feel very uncomfortable with.

1) The work is still largely placed in a host – parasite framework. While I can follow the reasoning for doing so (positive – negative interactions), I think it is misleading and distorts the picture. It needs a lot of explanation to understand why this system might be explained by a host – parasite metaphor. Reducing this to positive – negative interactions is rather simple and not informative. Furthermore, host – parasite interactions and mutualism are concepts invented to describe pairwise interactions. As soon as three or more players act together these concepts do not work anymore (see the debate on this in the host – microbiota literature). While I see that other worker in the field have also used the host-parasite framework, it is not commonly done so.

2) The treatment of temperature in the experiment is odd. Temperature is clearly a continuous variable, which in the experiment is broken into 3 categories. But the idea behind it is not an idea of categories, but rather of a continuum. This it is how it is presented in the interpretation of the study.

Reviewer #3:

The authors have done a nice job with the revision. We had a key disagreement, which they supported admirably, but I try to provide a bit more explanation so that this concern can be distinguished from picky semantics. The mite is not a parasite.

All deference to Tara Stewart's paper aside, figure 1 supplement does not solve my concern. I still don't agree that this study can be easily summarized as a host parasite relationship. I say this as someone that has explicitly defined such relationships in papers (see Lafferty and Kuris, 2002) and am weirdly troubled when other authors have not done so carefully. So, unfortunately, I have a strong opinion about this. And although the semantics might not be particularly important to most people, let me explain why they are to me in this particular case. It is a fascinating question how species interactions can change from positive to negative based on environmental conditions. And one often reads the assumption that it is the case that parasitism often changes over to mutualism based on the environmental context. The first sentence of your abstract does so with force (Ecological transitions between parasitism and mutualism are relatively commonplace). However, when reading carefully, these examples are usually not describing parasites, by which I mean consumers that have an intimate, non-lethal, dependent long-term feeding relationship on a single host individual during a particular parasitic life stage. When defined explicitly as such, transitions to mutualism is not a general rule about the nature of parasitism. It is far more common in other types of species associations like the one described here. When we refer to other parasite-like interactions, we usually take care to preface this as brood parasitism, or parasitoidism, or epiphyte, etc. By which I mean, describing this interaction as parasitism muddies our view about parasitism rather than clarifies our understanding of species interactions. The solution is to simply not rely on parasite as a shorthand to describe this particular species interaction. Rather, just describe how the species interact. It is quite interesting on its own, and calling it parasitism does not do it service. Below is an example of the edited Abstract that accomplishes this.

"Ecological transitions in and out of mutualism are relatively commonplace though the causal agents driving such change remain poorly understood. Here we show that temperature stress modulates the harm threatened by a common enemy, and thereby induces a phoretic mite to become a protective mutualist. Our experiments focus on the interactions between the burying beetle Nicrophorus vespilloides, an associated mite species Poecilochirus carabi and their common enemy, blowflies, when all three species reproduce on the same small vertebrate carrion. We show that mites compete with burying beetles for food in the absence of blowflies, because they reduce beetle reproductive success. However, when blowflies breed on the carrion too, mites enhance beetle reproductive success by eating blowfly eggs. High densities of mites are especially effective at promoting beetle reproductive success at higher and lower natural ranges in temperature, when blowfly larvae are more potent rivals for the limited resources on the carcass."

Similar care in terminology make it easy to get rid of parasite and specify the actual biology of the system.

Lafferty and Kuris, (2002).

[Editors' note: further revisions were suggested prior to acceptance, as described below.]

Thank you for submitting your article "Temperature stress induces mites to help their carrion beetle hosts by eliminating rival blowflies" for consideration by *eLife*.

Your revised article has been evaluated by a Reviewing Editor in consultation with a statistical adviser, and we would like to ask you to make some additional revisions before final acceptance. For details, please see below.

The Reviewing Editor commented on your response to a point that had been raised during the previous round of review, namely the treatment of temperature in statistical analyses. This is a lightly edited version of their feedback:

There are three types of independent variables: continuous variables (you know the order and the distance between the values: e.g., 1, 2, 3, 4…); ordered variables (you know the order (e.g., A > B > C), but you may not know the distance between the values); and factors (where the order does not play any role).

Temperature is continuous. The difference between 10C and 15C is the same as the difference between 15C and 20C. The authors prefer not to treat it as continuous, because they argue that, if you have treatments of different temperatures, you cannot know what happens in between these temperatures. This is right in a world where even very small temperature differences make big differences in the outcome. I don't think this is likely here, but this is not worth arguing about.

The next option would be to use ordered values for the temperature treatments (12{degree sign}C < 16{degree sign}C < 20{degree sign}C, disregarding how big the differences are). This allows you to say that 16{degree sign}C is in between 12{degree sign}C and 20{degree sign}C, and that 12{degree sign}C and 20{degree sign}C are low and high, respectively. This is what the authors use when they talk about "low", "intermediate" and "high" temperatures. Using these attributes makes only sense if the treatments are at least ordered, as otherwise you use a statistical model assuming no order, but construct arguments based on an order.

Why does this matter?

Statistically: Using temperature as a factor means that the variation of the independent variable is totally unconstrained. Treatments can vary in any way from each other. Using temperature as an ordered or continuous variable imposes a constraint on the way how the variation of the independent variable is structured. This reduces the power of the statistical test, but gives you more information as you understand more about the structure of the variance.

Biologically: A factor does not assume that the different treatments have anything to do with each other. An ordered or continuous variable assumes that treatments are not independent. This is what we can use in the argumentation. For example, when we talk about "intermediate" temperature, we can make a biological statement in relation to the more extreme treatments. We gain this freedom by a penalty in the statistical power.

The authors also argue that having just three values is not enough for a regression, which is incorrect. There are many independent replicates for each of the three temperatures, so it is not three but many. It is in fact possible to do a regression with just two temperatures (but multiple measurements per temperature). The aim of a regression is to estimate the change in the dependent variable in relation to the independent variable. This is exactly what was aimed for here.

Following further discussion, we decided to consult a statistical adviser on this matter. This is a lightly edited version of their feedback:

It is of course possible to treat the temperature variable as continuous even though the experiment has constrained it to be one of three specific values. The only reason that would be justified for treating it as a factor would be because they suspect there is a non-linear relationship between temperature and the outcome. This appears to be the case in some of their field experiments. So, while some of their defence is incorrect, their point about suspecting a U-shaped or non-linear relationship is justification for treating temperature as a categorical factor. If they have specific hypotheses (after the field observations and before the lab experiments) about low and high temperatures that would also make sense to treat as a factor.

We would like to ask you to address these considerations in your final revisions, which we feel can be achieved by clarifying the rationale of your approach to statistical analyses, and by using more nuanced wording for describing the findings from the experiments with three temperature treatments.

---

## [Author Response]

The three reviewers have raised a variety of issues concerning the presentation of the study. Here is a short summary, and further details are provided in the three separate reports appended below.1) Terminology. The general presentation of the work needs some revision. The reviewers made suggestions regarding the title and various terminological issues.

We have amended the terminology where it caused problems, throughout.

2) Framework. The placement of the work in a parasitism-mutualism framework was strongly criticised. The study would not lose any appeal if it was instead placed in a more conventional ecological framework but would gain in clarity.

We have now explained in great detail why we chose to use a parasitism-mutualism framework to analyse this study system. We have also incorporated the excellent ecological concepts raised by the reviewers into this way of thinking. In our view this strengthens the study by retaining the emphasis on fitness outcomes (required by the parasitism-mutualism framework) and blending it with a strong ecological perspective that explains how these interactions fit into a wider web of direct and indirect interactions.

3) Statistical issues. The Akaike Information Criterion is central for distinguishing between different statistical models. It is mentioned in the text but apparently not applied.

We have provided these details in the text.

4) Literature cited. In connection with point 2 (above), it will be necessary to cover relevant literature when placing the work in a food-web theoretical framework.

We have expanded our citation of the literature.

These issues, including those listed below, should be taken care of in a revised version.Reviewer #1:1) The title starts with "A changing climate…". This is far-fetched. The term climate shows only once up in the entire text (at the very end of the Discussion section). This title seems more aiming to catch attention than to give an adequate title for the study. I find this misleading and suggest changing this.

We agree and we have rewritten the impact statement (we assume this is what the reviewer is referring to here, although we have also rewritten the title).

2) The parasitism – mutualism framework is a far fetch. The authors justify their claim by citing a dictionary entry (supplement) about the term parasitism. Even so this entry is ok, it is not how it is perceived in the general community and I think therefore it is misleading. I don't understand why it has to be parasitism. The message of the study remains the same if one states that species interactions change sign (from positive to negative) when the environmental conditions change. The main interaction is that mites compete with the beetles for food, which is competition. Adding the blow flies to the picture turns the mites into predators of blow fly eggs. This is helpful for the host. So, it is a change from a competitor to a mutualist.

Thank you for your thoughts on this. As you have realised, it is not easy to define the interactions between burying beetles and their mites. Nevertheless, we have given this considerable thought. Furthermore, we are not the first to describe these mites as parasitic. However, since the terminology we have used has caused problems with more than one reviewer we decided to go back to the literature to more clearly explain the reasoning of our thinking.

Stewart and Schnitzer, (2017) set out the key differences between competition and parasitism, specifically focusing on interactions where the difference between them has become somewhat blurred. We follow their framework for defining mites as parasites rather than competitors and explain that using their conceptual roadmap in Figure 1—figure supplement 1.

The main reason for defining mites as parasites and not competitors lies in the effect that the interaction has on the fitness of mites versus the fitness of burying beetles: parasitism is a positive-negative interaction, whereas competition is a negative-negative interaction.

Mite fitness is positively affected by beetles because mites depend solely on beetles to be transported to their breeding resource (Schwarz and Müller, 1992), and because they then produce more offspring in the presence of beetles than in the absence of beetles (Sun and Kilner, 2019). Burying beetles, on the other hand, lose fitness to mites because mites compete with beetle adults and larvae for carrion resource (Nehring et al., 2017), and because mites can predate directly upon beetle eggs and larvae (Beninger, 1993; De Gasperin and Kilner, 2015). Therefore, mites are parasites of burying beetles.

We have now explained this thinking in the paper (Results section). It is also evident in Figure 1—figure supplement 1, which was also based on Stewart and Schnitzer, (2017) definitions.

3) Statistics: The authors show the models in the legends to the tables. But this is not complete. What are the different items in the model? Which of the items are factors, which are continuous variables? In the text some info is given, but is not very clear. What is a "categorical covariates". Is this a factor (like presence/absence of blow flies) or is this a continuous covariable (like temperature and carcass mass)? Please clarify. Temperature should be used throughout as a contiguous covariable, not as a factor. Furthermore, it would be helpful if the error distribution is given for each model. Given that the data have rather special error distributions (see figures) this is very important.

Thank you for this suggestion. We agree that our use of categorical covariates is misleading. We have now clarified variables in all statistical analyses sections, either as categorical factors or as continuous variables.

We included temperature firstly as a continuous variable for field experiment because we measured continuous natural variation in temperature. However, for laboratory experiments, we manipulated temperature by establishing three temperature treatment categories (i.e. 11, 15, and 19°C). This is why temperature is a categorical factor in the analyses of laboratory-collected data.

We describe the error distribution we used with these models (negative binomial distributions).

4) The authors state that they used "Akaike Information Criterion (AIC) by evaluating AICc, δ AICc and Akaike weights". I did not find any evidence in the statistics that this was actually done. This is important in particular for the fitting of the non-linear relationships.

We have now incorporated this information.

Reviewer #2:This study is very timely and unique, as it utilizes a combination of both field and lab work to examine protective mutualism using eukaryotes. A few comments:1) "Transition", which is used throughout the manuscript can refer to evolutionary history/trajectory or an ecological-time, context-dependent switch. Given the combination of references used herein on evolutionary and ecological transitions, we strongly recommend clarifying throughout and choosing references/wording more carefully. There are numerous references to condition-dependent transitions along the parasite-mutualist continuum across the tree of life which are not mentioned.

Thank you – this is a truly helpful suggestion. We have now made it clear throughout that transition in this study specifically refers to ecological transition throughout the manuscript.

2) There was also a lack of discussion of protective symbioses against abiotic stresses (like temperature) in the Introduction, which are different than the biotic stresses.Here are some references on protective microbes against abiotic stresses:Corbin et al., 2017 This reference was cited, but not sure the point of this paper was how it was used in the manuscript (–Introduction, "Yet theoretical analyses that consider how such interactions evolve and persist derive mainly from recent interest, with a particular focus on the microbial endosymbionts that can be induced to defend their hosts from attack"). This is a good source for examples of symbionts that protect hosts from temperature stresses and how temperature affects the association, not about biotic factors, which is what the sentence in the manuscript implies.Engl et al., 2018.Feldhaar, 2011.Hoang, Gerardo and Morran, 2019.

We have made the distinction between abiotic and biotic factors clearer throughout and also cited Engl et al., 2018 and Hoang et al., 2019 (Introduction).

3) Introduction is a bit sparse. For example, before the last paragraph, a brief sentence or two to summarize the big picture and tie to what is known to what is not known. Moreover, there was no mention of mutualist/symbiont density or mite density in the Introduction, even though that was what was driving the results

We agree with you that more details are needed to briefly summarise the bigger picture etc. We have re-written the Introduction to make that clearer.

We have also included a paragraph in the Discussion section to highlight how mite densities are linked to the transition from parasitism to mutualism.

4) A diagram could be used to show how each player interacts with one another, especially because there are several definitions being used and different environmental contexts involved. For example, in the Introduction: being a mutualist is not mutually exclusive of being predatory. Mites are mutualistic towards the beetles and predatory towards the blowflies. A diagram can represent this. Moreover, doesn't the definition of parasitism have to be in a supplement (seems like an odd place) as opposed to just mentioned when the system is introduced?

This is another great suggestion. We have made a summary diagram of our results, following the referee’s suggestions here (Figure 5). Our use of the term ‘parasitism’ has turned out to be more controversial than we expected. We now explain the logic behind using more fully in the Introduction and in Figure 1—figure supplement 1.

5) Another point to mention in the Discussion section would be how the association becomes more mutualistic when there are more mites: the results indicate that more mites increase beetle fitness, which in turn benefit the mites (at the least, in order for beetle fitness to increase there has to be more mites. I'm not saying that the beetles directly promote mite fitness, but there is an association there. Also, perhaps more beetles could mean more mites can hitchhike?).

We agree that we should have mentioned in the Discussion section the effect of mite density on any movement along the parasitism-mutualism continuum. We have now made that clear.

The effects of beetle success on mite success are more complicated to discern than perhaps the referee realizes. In the field, most mite offspring disperse away on the adult beetles (Schwarz and Müller, 1992), rather than on beetle larvae produced during reproduction (as explained in the Results section). Mites are mostly transmitted horizontally between adults. So, while it is true that more beetles means more hosts for mites, it hard to link reproductive success at each breeding event to the subsequent reproductive success of the mites produced from the same carcass.

6) How common is it for phoresy to be parasitic? Worth mentioning how your consideration of the costs of mites squares with phoresy as currently understood in the literature.

Since phoretic interactions remain relatively understudied, it is hard to give an accurate answer to this question (White et al., 2017). We have followed your advice, though, and given two examples in the Introduction to show that burying beetle mites are not unusual in this respect (Results section). The main observation is that the ‘cost-free’ phoretic transport phase paves the way for a continued intimate association, after the host and phoront arrive at their destination. At this point, the relationship can become more parasitic (or more mutualistic).

7) Supplementary file 1, Supplementary file 2, Supplementary file 3 and Supplementary file 4, but particularly Supplementary file 3. Given the size of these models, do the authors have enough power to detect significant differences?

This is a good point. We have now performed power analyses for the mixed models presented in Supplementary file 3. We found all of these models were sufficient to reject the null hypotheses, with the 97%, 97%, and 98.2% power, for analyses of burying beetle larvae, blowfly larvae, and mite offspring, respectively. We have also performed power analyses for Supplementary file 1, Supplementary file 2, and Supplementary file 4. All these results are now included in the table captions. These powers were calculated using simulations based on 1000 Monte Carlo simulations, with the function powerSim in the package *SIMR*.

Reviewer #3:[…] The paper is relatively well written and the authors are to be commended for their use of manipulative field and laboratory experiments. I enjoyed reading the paper and learned a lot. I do, however, have several suggestions that I think will improve the paper.The title is vague. Authors sometimes assume that this will help the paper appear general and therefore capture broader readership. It doesn't. At least I would not bother to read it based on the title. And I think you want people like me to read it. A good title should state the key results and identify the system. I would read a paper titled something like: Phoretic mites help their carrion beetle hosts under temperature stress by eating competing blowfly larvae.

We have changed the title to make it more specific:

Temperature stress induces mites to help their carrion beetle hosts by eliminating rival blowflies

A general point about the Introduction: the paper will flow better (and read more like a scientific paper) if the authors clearly state the series of predictions they are going to test an how these predictions flow from their hypotheses. Each Pvalue in the result should be an evaluation of either an apriori prediction or the assessment of an assumption, or a controlling variable. And it should be clear how the hypotheses derive from the literature, theory or field observations.

We have rewritten the Introduction to make it clearer how we arrived at the questions we investigated in this study. As is evident from the experimental design, we expected that blowflies, temperature and mites – either alone or in combination – might influence burying beetle reproductive success. We have also restructured the Discussion section about the same set of questions.

The paper is missing some key information on what happens in nature. In particular, what is the natural distribution of mites on beetles in the field (this will help readers understand whether the treatments in the experiments are reasonable).

We have now provided this information by referring to our previously published paper (Sun et al., 2019), which describes the frequency distribution of mite density per beetle in their natural habitats.

This gets me to my point about whether the mite is a parasite or a mutualist or a whatever. This might be semantics, but I put a high bar on assigning parasitism. And the beetle-mite interaction does not cross that bar for me. Namely, the negative trophic interaction does not appear to occur during intimacy.

We can see we should have explained the natural history more clearly.

For example, we disagree on the point of about ‘intimacy’. The burying beetles and mites remain intimately associated during reproduction on the carcass. Mite offspring and larvae are in sufficiently close proximity that they bump into each other frequently during this period. We explain this more clearly now in the text (Results section).

Parasitism must be defined at the stage level rather than the species level.

Again we disagree – mainly because we don’t understand what additional conceptual insights we can gain by doing this, especially as we are working with a definition of parasitism that is centred upon fitness (and this makes it hard to separate parents from offspring conceptually). It’s also impossible to disentangle the stages in a practical sense. The parents and offspring generations of both mites and beetles mix together on the carcass, for example. The mite offspring are also dependent on the beetle parents to disperse away from the breeding site at the end of reproduction.

I also put a particularly high bar for deciding that parasitism is context dependent when simpler explanations via indirect effects are possible. To me it is clear that the phoretic mite stage is not a parasite because it does not feed on the beetle and the free-living mite stage is not a parasite because it neither feeds on the beetle or is intimate with the beetle.

We explain why we disagree with this characterization of the ‘free-living stage’ above. We now clarify that mites are mildly parasitic on beetles because they can directly predate on beetle eggs and larvae (Beninger, 1993; De Gasperin and Kilner, 2015; Nehring et al., 2017; Sun et al., 2019) (Results section). Therefore, mites can reduce beetle fitness both by direct predation as well as by indirect competition for carrion resource.

This does not mean the interaction is not interesting. In fact, I think it is far more interesting than the way the authors have packaged it. The mite is a phoretic commensal that uses its host to be transported to a common food source. Once on that common food source, the relationship can become competitive. But that competition can lead to a net indirect benefit under situations where another competitor (blowflies) might dominate. This interpretation is better couched in food-web theory rather than evolutionary theory.

We are grateful to the referee for pointing out a food web way of thinking about the results. We have integrated this style of thinking into the Discussion now and in a new figure to summarise the results (Figure 5).

As an example, here is a paper I remembered from decades ago that pondered over the relationship between mites and hosts and third-party interactions. I include the citation for the authors' interest. This author had similar observations but did not conclude a plastic parasitic-mutualistic relationship.Rigby, (1996).

We now cite a paper by Stewart and Schnitzer, (2017), which clarifies the difference between competition and parasitism (summarized in Figure 1—figure supplement 1), and upon which we base the framing of the Introduction.

[Editors' note: further revisions were suggested prior to acceptance, as described below.]

The reviewers were mostly satisfied with your responses to their comments from the first round of review. However, there are still points that require further attention. In particular, both reviewers remained unconvinced by your assertion that this is a host-parasite system. We suggest the following course of action for addressing this concern: We feel there is no need to stress the host-parasite framework in the abstract, introduction, and result section – the work is very interesting without this. If you wish, you can add a short paragraph to the Discussion section that outlines the reasons why you think this system can also be viewed productively within a host-parasite framework. In this section, if you choose to include it, please contrast your interpretation with alternative views, as presented by the reviewers – that way, readers can hear both sides of the argument.Further comments from reviewers #1 and #3 follow.Reviewer #1:The revised manuscript fixed many of the issues I had with the earlier version. The authors did a nice job. There are still two points I feel very uncomfortable with.1) The work is still largely placed in a host – parasite framework. While I can follow the reasoning for doing so (positive – negative interactions), I think it is misleading and distorts the picture. It needs a lot of explanation to understand why this system might be explained by a host – parasite metaphor. Reducing this to positive – negative interactions is rather simple and not informative. Furthermore, host – parasite interactions and mutualism are concepts invented to describe pairwise interactions. As soon as three or more players act together these concepts do not work anymore (see the debate on this in the host – microbiota literature). While I see that other worker in the field have also used the host-parasite framework, it is not commonly done so.

We accept your point and we have now framed mite interactions as negative relationships instead of parasitism, wherever is relevant.

2) The treatment of temperature in the experiment is odd. Temperature is clearly a continuous variable, which in the experiment is broken into 3 categories. But the idea behind it is not an idea of categories, but rather of a continuum. This it is how it is presented in the interpretation of the study.

We agree with you that temperature is a continuously varying trait in nature, and this is why we analysed the data by including temperature as a continuous variable in our field experiment. However, in the lab we did not allow temperature to vary continuously in our experimental design, because it was easier to manipulate temperature by creating experimental categories of low, medium and high – within the natural range.

We use this approach commonly in our experimental designs. We take traits that vary continuously in nature and we subject them to experimental analysis by creating treatments in which variation (within the natural range) is cast into categories, so that we can more easily disentangle cause and effect. Here we have used the same approach to understand temperature by creating three treatments 11, 15, and 19°C. We did not measure temperature within each incubator so have no further information about temperature within each treatment.

In our experimental datasets, these variables of interest no longer vary continuously, but instead are made to resemble factors by our experimental design. Thus 13°C and 17°C do not exist in our experiments. This is why we have analysed temperature as a factor. We cannot draw inferences about effects at e.g. 13°C and 17°C in the lab because we did not measure them (unlike with the field dataset).

Furthermore, since there are only three different quantitative variables (11, 15, and 19°C), fitting a line or curve through the data is not appropriate.

Finally, we have good reasons (from our field data) for supposing that the effects of temperature are non-linear, which makes the business of drawing inferences about effects at temperatures we did not evaluate even trickier.

Reviewer #3:The authors have done a nice job with the revision. We had a key disagreement, which they supported admirably, but I try to provide a bit more explanation so that this concern can be distinguished from picky semantics. The mite is not a parasite.All deference to Tara Stewart's paper, figure 1 supplement does not solve my concern. I still don't agree that this study can be easily summarized as a host parasite relationship […] Below is an example of the edited Abstract that accomplishes this."Ecological transitions in and out of mutualism are relatively commonplace though the causal agents driving such change remain poorly understood. […] High densities of mites are especially effective at promoting beetle reproductive success at higher and lower natural ranges in temperature, when blowfly larvae are more potent rivals for the limited resources on the carcass."Similar care in terminology make it easy to get rid of parasite and specify the actual biology of the system.Lafferty and Kuris, (2002).

Thank you – this is very helpful indeed. We have now rephrased interactions between burying beetles and mites as negative relationships instead of parasitisms throughout the manuscript. We would also like to thank you for updating our abstract.

[Editors' note: further revisions were suggested prior to acceptance, as described below.]

The Reviewing Editor commented on your response to a point that had been raised during the previous round of review, namely the treatment of temperature in statistical analyses. This is a lightly edited version of their feedback: […]We would like to ask you to address these considerations in your final revisions, which we feel can be achieved by clarifying the rationale of your approach to statistical analyses, and by using more nuanced wording for describing the findings from the experiments with three temperature treatments.

Thank you very much again for your advice on the use of temperature as a categorical factor. We had a good *a priori* reason for suspecting a non-linear effect of temperature, from the data we collected with our field experiment (Figure 1). This is (partly) why we adopted the conservative approach of including temperature as a categorical variable. We have now made that clear in the description of the statistical models in the Materials and methods section.

We have also changed the description of the results to add nuance (Results section, Discussion section).